# Genetic variation in *POT1* and risk of thyroid subsequent malignant neoplasm: A report from the Childhood Cancer Survivor Study

**Melissa A. Richard**[1], **Philip J. Lupo**[1], **Lindsay M. Morton**[2], **Yutaka A. Yasui**[3], **Yadav A. Sapkota**[3], **Michael A. Arnold**[4], **Geraldine Aubert**[5], **Joseph P. Neglia**[6], **Lucie M. Turcotte**[6], **Wendy M. Leisenring**[7], **Joshua N. Sampson**[2], **Stephen J. Chanock**[2,8], **Melissa M. Hudson**[9], **Gregory T. Armstrong**[3], **Leslie L. Robison**[3], **Smita Bhatia**[10], **Maria Monica Gramatges**[1] *

**1** Department of Pediatrics, Baylor College of Medicine and Dan L. Duncan Cancer Center, Houston, TX, United States of America, **2** Division of Cancer Epidemiology and Genetics, National Cancer Institute, Bethesda, MD, United States of America, **3** Department of Epidemiology and Cancer Control, St. Jude Children's Research Hospital, Memphis, TN, United States of America, **4** Department of Pathology and Laboratory Medicine, Nationwide Children's Hospital, Columbus, Ohio, United States of America, **5** Terry Fox Laboratory, British Columbia Cancer Agency, Vancouver, BC, Canada, **6** Department of Pediatrics, University of Minnesota, Minneapolis, MN, United States of America, **7** Division of Clinical Research, Fred Hutchinson Cancer Research Center, Seattle, WA, United States of America, **8** Cancer Genomics Research Laboratory, Leidos Biomedical Research, Inc., Frederick National Laboratory for Cancer Research, Frederick, Maryland, United States of America, **9** Department of Oncology, St. Jude Children's Research Hospital, Memphis, TN, United States of America, **10** Institute for Cancer Outcomes and Survivorship, University of Alabama at Birmingham, Birmingham, AL, England

* gramatge@bcm.edu

## Abstract

### Background

Telomere length is associated with risk for thyroid subsequent malignant neoplasm in survivors of childhood cancer. Here, we investigated associations between thyroid subsequent malignant neoplasm and inherited variation in telomere maintenance genes.

### Methods

We used RegulomeDB to annotate the functional impact of variants mapping to 14 telomere maintenance genes among 5,066 five-or-more year survivors who participate in the Childhood Cancer Survivor Study (CCSS) and who are longitudinally followed for incidence of subsequent cancers. Hazard ratios for thyroid subsequent malignant neoplasm were calculated for 60 putatively functional variants with minor allele frequency $\geq$1% in or near telomere maintenance genes. Functional impact was further assessed by measuring telomere length in leukocyte subsets.

### Results

The minor allele at Protection of Telomeres-1 (*POT1)* rs58722976 was associated with increased risk for thyroid subsequent malignant neoplasm (adjusted HR = 6.1, 95% CI: 2.4, 15.5, P = 0.0001; Fisher's exact P = 0.001). This imputed SNP was present in three out of 110 survivors who developed thyroid cancer vs. 14 out of 4,956 survivors who did not

**Data Availability Statement:** The CCSS genome-wide association study (GWAS) dataset is available to investigate the role of genetic susceptibility in

the development of malignant and non-malignant treatment-related outcomes in cancer survivors. This process is open to investigators through collaboration with CCSS and National Cancer Institute investigators in the use of existing GWAS data and corresponding outcomes-related data to address innovative research questions relating to potential genetic contributions to risk for treatment-related outcomes through submission of an Application of Intent (https://ccss.stjude.org/develop-a-study/gwas-data-resource.html). Genotype data for the CCSS are also available through dbGaP, accession phs001327.v1.p1.

**Funding:** This work was supported by the National Cancer Institute (NCI) (CA194473: M.M.G., Principal Investigator) and a CCSS Career Development Award to P.J.L. CCSS is supported by the NCI (CA55727: G.T.A., Principal Investigator). Genotyping for CCSS was supported by the Intramural Research Program of the NCI, National Institutes of Health. The sponsors played no role in study design, data collection, analysis, decision to publish, or manuscript preparation.

**Competing interests:** I have read the journal's policy and the authors of this manuscript have the following competing interests: Geraldine Aubert is a paid employee of Repeat Diagnostics, Inc., which conducted the telomere length assessment. we can affirm that the employment of Dr. Aubert does not alter our adherence to PLOS ONE policies on sharing data and materials.

develop thyroid cancer. In a subset of 83 survivors with leukocyte telomere length data available, this variant was associated with longer telomeres in B lymphocytes (P = 0.004).

## Conclusions

Using a functional variant approach, we identified and confirmed an association between a low frequency intronic regulatory *POT1* variant and thyroid subsequent malignant neoplasm in survivors of childhood cancer. These results suggest that intronic variation in *POT1* may affect key protein binding interactions that impact telomere maintenance and genomic integrity.

## Introduction

Over 80% of individuals diagnosed and treated for cancer as children will survive five or more years after completing cancer treatment. An increased risk for subsequent malignant neoplasms (SMN), including SMNs of the thyroid (thyroid SMN), is a known late effect of childhood cancer treatment.[1] Higher risk for thyroid SMN is observed among female survivors, those diagnosed with primary cancer at a younger age, and those exposed to radiation and certain chemotherapeutic agents.[2] However, these clinical risk factors do not fully explain variability in thyroid SMN risk, suggesting a likely role for inherited genetic factors.

Telomeres are repetitive DNA-protein structures localized to chromosome ends, protecting chromosome integrity and loss of proximal terminal coding regions during DNA replication. Telomere length is determined by environmental and hereditary factors, shortens with age, and is maintained by telomerase and associated proteins. The majority of population-based studies have noted an inverse association between cancer risk over time and leukocyte telomere length when measured directly through standard methodologies.[3, 4] However, data from genome wide association studies suggest that longer telomere length may also confer risk for a number of cancers.[5] This apparent paradox is hypothesized to result from multi-stage mechanisms underlying malignant transformation: specifically, accumulation of random mutational events during stem cell replication may lead to an increase in sporadic cancer risk that occurs with physiologic aging, for which telomere shortening is a proxy. Longer telomeres confer a greater capacity for cellular clonal expansion and proliferation, so that individuals with very long telomeres may also be at especially high risk for carcinogenesis.[6]

Exposure to ionizing radiation induces DNA damage and may lead to telomere dysfunction,[7–10] providing rationale to suggest a relationship between telomere length and risk for SMN among radiation-exposed cancer survivors. We previously reported an increased risk for thyroid SMN in radiation-exposed survivors of childhood cancer with reduced leukocyte telomere content.[11] We subsequently observed no association between genotypically-estimated telomere length, determined from variation in nine common telomere length-associated SNPs, and thyroid SMN.[12] These prior works prompted a functional variant approach to further interrogate the relationship between variation in genes related to telomere maintenance and thyroid SMN in the Childhood Cancer Survivor Study (CCSS).

## Materials and methods

### Subjects

The CCSS is a multi-center cohort of individuals diagnosed <21 years of age with childhood cancer between 1970 and 1986, and who survived five or more years after completion of cancer

treatment.[13] After enrollment to the CCSS, survivors are prospectively followed through self-report questionnaires to ascertain late effects of cancer treatment. Thyroid SMN was defined as any SMN of the thyroid gland occurring as the first subsequent neoplasm in a CCSS participant, with a diagnosis and date of diagnosis that had been verified from the original pathology report by a CCSS-designated pathologist. All subjects provided written consent to participate in the CCSS, and each of the 26 participating institutions obtained approval to conduct this research through their institutional IRB. Diagnosis and treatment data were initially abstracted from the medical record by the participating treating institution and submitted, fully anonymized, to CCSS. The work described in this study utilized fully anonymized data. The study was conducted in accordance with the Declarations of Helsinki.

## Genetic data

This study leveraged genetic data from 5,066 CCSS participants with complete follow-up for SMN. DNA was extracted using standard methods from blood, saliva (Oragene), or buccal cells (mouthwash), collected at least five years from diagnosis and genotyped using the Illumina HumanOmni5Exome array at the Cancer Genomics Research Laboratory of the National Cancer Institute. All survivors were imputed to the 1000 Genomes reference haplotypes.[14] We mapped 3,499 variants to a 100 bp region flanking genes implicated in a telomere biology disorder: *ACD*, *CTC1*, *DKC1*, *NAF1*, *NHP2*, *NOP10*, *PARN*, *POT1*, *RTEL1*, *STN1*, *TERC*, *TERT*, *TINF2*, *WRAP53* (**Table 1**).[15–31] We restricted our analyses to include only functional SNPs with minor allele frequency (MAF) $\geq$ 1% in or near these 14 genes that were considered most likely to affect transcriptional factor binding, defined by a RegulomeDB score $\leq$ 2, which signifies localization to transcriptional factor binding and motifs, DNase footprints and peaks, or identification as a quantitative trait locus for gene expression (eQTL)

**Table 1. Genes currently implicated in telomere biology disorders.**

| Gene name (HGNC[a] Symbol) | Related telomere biology disorder(s)[b] | Role in telomere maintenance |
|---|---|---|
| *ACD* | HHS, AA, familial cancers | Part of the shelterin complex |
| *CTC1* | DC, Coats Plus, cerebroretinal microangiopathy | Part of the CST complex |
| *DKC1* | DC, HHS | Part of the telomerase holoenzyme |
| *NAF1* | PF | Part of the telomerase holoenzyme |
| *NHP2* | DC | Part of the telomerase holoenzyme |
| *NOP10* | DC | Part of the telomerase holoenzyme |
| *PARN* | DC, PF, HHS | Ribonuclease interacting with TERC |
| *POT1* | Coats Plus, familial cancers | Part of the shelterin complex |
| *RTEL1* | DC, PF, HHS | DNA helicase interacting with shelterin |
| *STN1 (OBFC1)* | Coats Plus | Part of the CST complex |
| *TERC* | DC, PF, MDS, HHS, AA | Part of the telomerase holoenzyme |
| *TERT* | DC, PF, AML, MDS, HHS, AA | Part of the telomerase holoenzyme |
| *TINF2* | DC, HHS, RS, AA | Part of the shelterin complex |
| *WRAP53* | DC | Protein that binds to TERC |

[a]HUGO Gene Nomenclature Committee

[b]HHS = Hoyeraal Hreidersson Syndrome, DC = dyskeratosis congenita, PF = pulmonary fibrosis,

MDS = myelodysplastic syndrome, AA = aplastic anemia, AML = acute myeloid leukemia, RS = Revesz Syndrome

across multiple tissues.[32] Genetic variants were coded as imputed genotype dosages and filtered for imputation quality >0.7.

## Statistical analysis

We conducted time-to-event Cox regression for thyroid SMN as a first SMN using the *survival* package in R v3.5.2. The at-risk period began with the date of initial cancer diagnosis and ended at the date of thyroid SMN diagnosis, or the earliest first report of other SMN, death, and/or date of last follow up. Relative risk of thyroid SMN was estimated using hazard ratios (HR), adjusted for demographic and clinical factors including sex, birth year before or after 1970, age at primary cancer diagnosis, primary cancer diagnosis, radiation exposure (yes/no), neck radiation exposure (yes/no), alkylating agent exposure (yes/no), and thyroid nodules (yes/no). Genetic ancestry proportions were estimated using three continental ancestries (CEU, AFR, and ASN) in STRUCTURE.[33] We used an 80% threshold to define individuals of a predominant ancestry and performed secondary analyses 1) in the total sample additionally adjusted for estimated European and African proportions and 2) restricted to individuals of European ancestry. Statistical significance for association with risk for thyroid SMN was defined by the Bonferroni correction for the number of variants tested ($\alpha$ = 0.00083). For lower frequency variants (MAF 1–5%) that were statistically significant in the Cox regression model, we validated the regression model using a Fisher's exact test to evaluate differences in allele frequency between those with and without thyroid SMN.[34]

## Measurement of telomere length in hematopoietic cells

Viably frozen leukocyte samples were available for 83 CCSS subjects included in this study. Leukocyte telomere length was measured by flow cytometry fluorescence in situ hybridization (flow FISH) following established procedures.[35] Briefly, leukocyte telomere length was assessed against that of control bovine thymocytes after denaturation in formamide at 87˚C. Quantitative hybridization with a fluorescein-conjugated $(CCCTAA)^3$ peptide nucleic acid (PNA) probe specific for telomere repeats (in-house synthesis) was then performed and counterstained with LDS751 DNA dye (Exciton), followed by analysis with flow cytometry. Results were transformed to age-adjusted percentiles based on the date of sample collection. For significant functional variants, we compared the proportions of age-based relative telomere length categories (very low, low, normal, high, or very high) by cell type between carriers and non-carriers of the risk allele using Fisher's exact test.

## Results

We identified 110 CCSS participants who developed thyroid SMN five or more years after completion of cancer treatment and 4,956 survivors without thyroid SMN. Survivors who developed thyroid SMN were more likely to be female (thyroid SMN: 62.7%, non-cases: 51.6%) and to have thyroid nodules (thyroid SMN: 84.5%, non-cases: 10.0%). Thyroid SMN also occurred more frequently for those with older age at childhood cancer diagnosis (thyroid SMN cases: mean 9.0 years, non-cases: mean 7.9 years) and a primary diagnosis of Hodgkin lymphoma (thyroid SMN cases: 32.7%, non-cases: 12.6%). Among primary cancer treatment characteristics, radiation treatment to the neck and exposure to alkylating chemotherapy was also more likely to have occurred among thyroid SMN cases than survivors without thyroid SMN (**Table 2**).

There were 60 SNPs included in our analyses located in or near telomere candidate genes that had both 1) a RegulomeDB score ≤2 signifying high likelihood for affecting transcriptional regulation and 2) a general population MAF <1%. Only one imputed variant

**Table 2. Characteristics of the Childhood Cancer Survivor Study participants by development of subsequent malignant neoplasm of the thyroid.**

| | | Thyroid SMN cases n = 110 | | Non-cases n = 4,956 | |
|---|---|---|---|---|---|
| | | n (%) | | n (%) | |
| Age at first malignancy, years (mean ± SD) | | 9.0 | ± 5.5 | 7.9 | ± 5.9 |
| Year of birth | | | | | |
| | Before 1970 | 68 | 61.8% | 2,227 | 44.9% |
| | After 1970 | 42 | 38.2% | 2,729 | 55.1% |
| Sex | | | | | |
| | Male | 41 | 37.3% | 2,397 | 48.4% |
| | Female | 69 | 62.7% | 2,559 | 51.6% |
| Type of first malignancy | | | | | |
| | Leukemia | 30 | 27.3% | 1,589 | 32.1% |
| | Central nervous system | 12 | 10.9% | 591 | 11.9% |
| | Hodgkin lymphoma | 36 | 32.7% | 623 | 12.6% |
| | Non-Hodgkin lymphoma | 5 | 4.5% | 397 | 8.0% |
| | Kidney/Wilms tumor | 5 | 4.5% | 486 | 9.8% |
| | Neuroblastoma | 5 | 4.5% | 366 | 7.4% |
| | Soft tissue sarcoma | 6 | 5.5% | 463 | 9.3% |
| | Bone | 11 | 10.0% | 441 | 8.9% |
| Alkylating chemotherapy | | 69 | 62.7% | 2,542 | 51.3% |
| Any radiation treatment | | 94 | 85.5% | 3,144 | 63.4% |
| Radiation treatment to the neck | | 66 | 60.0% | 1,041 | 21.0% |
| Thyroid nodules | | 93 | 84.5% | 498 | 10.0% |

(imputation quality $r^2$ = 0.95) in an intronic region of *POT1* (Protection of Telomeres 1), rs58722976, met Bonferroni criteria for statistical significance (adjusted HR = 6.1, 95% CI: 2.4, 15.5, P = 0.0001). The risk-associated G allele was present in three individuals with thyroid SMN and 14 individuals without thyroid SMN (**Table 3**). Comparing the risk allele frequencies between cases and non-cases also supported association of rs58722976 with thyroid SMN (Fisher's exact P = 0.001). Estimated allele frequency at rs58722976 in the Genome Aggregation Database (gnomAD) suggests variation at rs58722976 occurs at highest frequency in individuals of African ancestry (AFR f(G) = 4.8%). In secondary analyses, we identified a consistent association of rs58722976 with thyroid SMN both when additionally adjusted for ancestry proportions (ancestry-adjusted HR = 8.0, 95% CI: 2.3, 27.2, p = 0.0009) and when restricted to European ancestry CCSS participants (CEU only HR = 18.9, 95% CI: 3.5, 101.7, p = 0.0006; median CEU proportion = 95.7%).

**Table 3. Genotype frequencies and Cox regression estimates for *POT1* rs58722976 and risk of subsequent malignant neoplasm of the thyroid in the Childhood Cancer Survivor Study.**

| | | Genotypes for thyroid SMN cases | | | Genotypes for non-cases | | | Cox regression estimates | | |
|---|---|---|---|---|---|---|---|---|---|---|
| SNP | Population | GG | AG | AA | GG | AG | AA | HR | 95% CI | P-value |
| rs58722976 | total sample | 1 | 2 | 107 | 0 | 14 | 4,942 | 6.1 | (2.4, 15.5) | 0.0001 |
| | AFR and CEU adjusted | 1 | 2 | 107 | 0 | 14 | 4,942 | 8.0 | (2.3, 27.2) | 0.0009 |
| | CEU only | 0 | 2 | 102 | 0 | 2 | 4,621 | 18.9 | (3.5, 101.7) | 0.0006 |

All models are adjusted for sex, age at primary cancer diagnosis, primary cancer diagnosis, decade of birth, and treatment exposures.

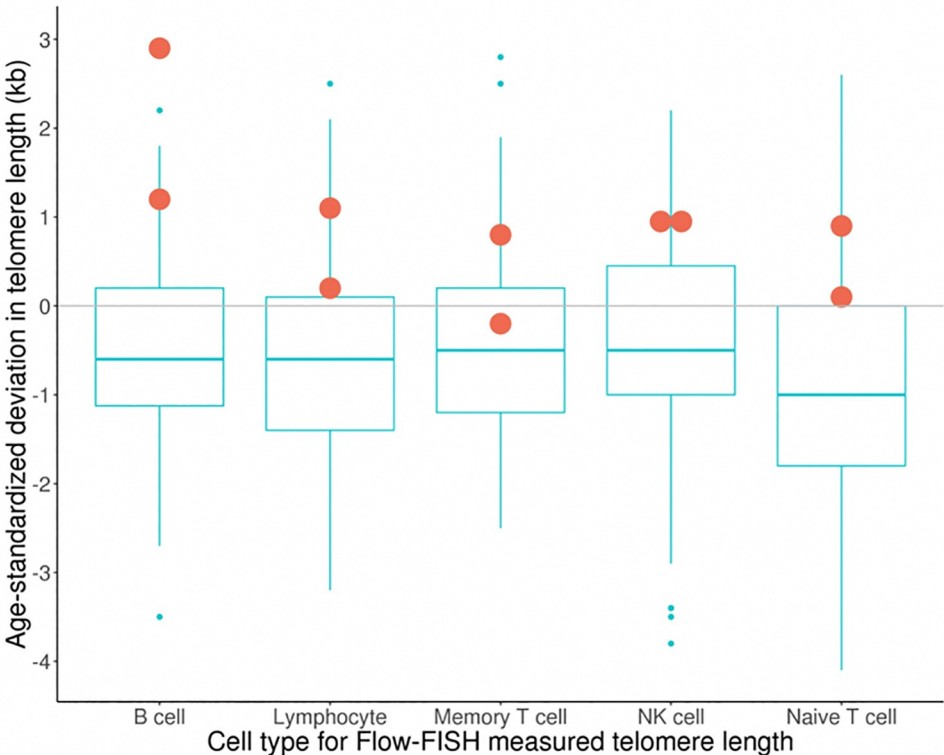

**Fig 1. Deviation from the age-based mean leukocyte telomere length measured by flow-FISH in survivors of childhood cancer.** Two of these survivors (one CEU and one YRI) were heterozygous for the high risk *POT1* rs58722976-G allele (denoted by circles) and demonstrate telomere length that is above the median for all leukocyte subsets and significantly longer among B lymphocytes than telomere length in survivors who lack this variant. Boxes include values falling between the 25th and 75th percentile deviation in telomere length from the age-based mean.

Lastly, telomere length was assessed in leukocyte populations in a subset of 83 survivors of childhood cancer enrolled to the CCSS. Only two were heterozygous for the risk allele in our top SNP. Telomere length was increased for the two heterozygous subjects across all leukocyte subsets compared with the median telomere length for subjects without the risk allele, a difference that was statistically significant for B lymphocytes (**Fig 1**, P = 0.004).

## Discussion

*POT1* is a highly conserved gene encoding a key component of the shelterin complex, which protects telomere ends against DNA damage recognition and facilitates telomerase-mediated telomere maintenance. *POT1* rs58722976 is an intronic variant identified by the ENCODE Consortium as a strong enhancer and DNase I hypersensitive site in multiple tissues, including the hematopoietic compartment, and may affect protein binding in components of the cohesion complex that play key roles in cancer etiology and maintaining genomic integrity.[36] Germline variants in *POT1* have been described in association with various cancer types[37] including familial glioma,[38] familial melanoma,[39–41] colorectal, ovarian, and lung cancer, [42] chronic lymphocytic leukemia,[43] multiple myeloma,[44] and non-*TP53* familial cancer syndromes.[45] Similar to our analyses, many of these genetic association studies note longer leukocyte telomere length among affected individuals compared with those who are unaffected.[38, 40, 41, 45] Recent data suggest that mutation-induced disruptions in the *POT1-TPP1* complex, both components of shelterin, affect the ability of this complex to bind

to telomeric DNA, leading to longer and more fragile telomeres that may promote genomic instability and cancer risk.[46]

This study was conducted within the CCSS, the largest genotyped population of survivors of childhood cancer. However, the low frequency of this variant precludes assessment of gene-environment interactions and adequately-sized genotyped survivor populations for replication or stratification among non-white ancestries. For example, all risk allele carriers excluded from CEU-only analysis were of primarily African ancestry (one homozygous individual with thyroid SMN and 12 carriers without thyroid SMN). Although thyroid cancer incidence is highest among individuals of European ancestry,[47] African ancestry confers a higher risk for the follicular variant of papillary thyroid cancer,[48] which was the SMN subtype observed in the survivor with thyroid SMN that was homozygous for the rs58722976 risk allele.

Using an approach that mapped functional variants to candidate genes, we identified an association between a low frequency intronic regulatory variant in *POT1* and risk for thyroid SMN in survivors of childhood cancer. We provide evidence that genetic variation at this locus may related to longer telomere length, in line with prior observations of longer leukocyte telomere length in association with cancers characterized by germline mutations in *POT1*. Our findings support a potential role for genetic variation in *POT1* affecting telomere maintenance and risk for thyroid SMN in survivors, suggesting the need for further study as larger genotyped survivor datasets emerge.

## Acknowledgments

The authors would like to thank the survivors and their families who participate in the Childhood Cancer Survivor Study.

## Author Contributions

**Conceptualization:** Philip J. Lupo, Smita Bhatia, Maria Monica Gramatges.

**Data curation:** Melissa A. Richard, Wendy M. Leisenring, Stephen J. Chanock, Gregory T. Armstrong, Leslie L. Robison, Smita Bhatia.

**Formal analysis:** Melissa A. Richard, Philip J. Lupo, Lindsay M. Morton, Yutaka A. Yasui, Yadav A. Sapkota, Geraldine Aubert, Wendy M. Leisenring, Joshua N. Sampson, Stephen J. Chanock.

**Funding acquisition:** Philip J. Lupo, Stephen J. Chanock, Gregory T. Armstrong, Maria Monica Gramatges.

**Investigation:** Melissa A. Richard, Philip J. Lupo, Lindsay M. Morton, Yutaka A. Yasui, Michael A. Arnold, Geraldine Aubert, Joseph P. Neglia, Lucie M. Turcotte, Wendy M. Leisenring, Joshua N. Sampson, Stephen J. Chanock, Melissa M. Hudson, Gregory T. Armstrong, Leslie L. Robison, Smita Bhatia, Maria Monica Gramatges.

**Methodology:** Melissa A. Richard, Philip J. Lupo, Lindsay M. Morton, Yutaka A. Yasui, Yadav A. Sapkota, Michael A. Arnold, Geraldine Aubert, Joseph P. Neglia, Lucie M. Turcotte, Joshua N. Sampson, Stephen J. Chanock, Melissa M. Hudson, Gregory T. Armstrong, Leslie L. Robison, Smita Bhatia, Maria Monica Gramatges.

**Project administration:** Yutaka A. Yasui, Michael A. Arnold, Joseph P. Neglia, Wendy M. Leisenring, Stephen J. Chanock, Melissa M. Hudson, Gregory T. Armstrong, Leslie L. Robison, Smita Bhatia, Maria Monica Gramatges.

**Resources:** Lindsay M. Morton, Yutaka A. Yasui, Wendy M. Leisenring, Gregory T. Armstrong, Leslie L. Robison, Smita Bhatia.

**Supervision:** Philip J. Lupo, Yutaka A. Yasui, Joseph P. Neglia, Stephen J. Chanock, Gregory T. Armstrong, Maria Monica Gramatges.

**Writing – original draft:** Maria Monica Gramatges.

**Writing – review & editing:** Melissa A. Richard, Philip J. Lupo, Lindsay M. Morton, Yutaka A. Yasui, Yadav A. Sapkota, Michael A. Arnold, Geraldine Aubert, Joseph P. Neglia, Lucie M. Turcotte, Wendy M. Leisenring, Joshua N. Sampson, Stephen J. Chanock, Melissa M. Hudson, Gregory T. Armstrong, Leslie L. Robison, Smita Bhatia, Maria Monica Gramatges.

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
