## [Decision Letter · Decision Letter 0]

10 Jan 2020

PONE-D-19-33962

Genetic variation in POT1 and risk of thyroid subsequent malignant neoplasm: A report from the Childhood Cancer Survivor Study

PLOS ONE

Dear Dr. Gramatges,

Thank you for submitting your manuscript to PLOS ONE. After careful consideration, we feel that it has merit but does not fully meet PLOS ONE’s publication criteria as it currently stands. Therefore, we invite you to submit a revised version of the manuscript that addresses the points raised during the review process.

This manuscript is both a well-designed study and a well-written manuscript.  However, we need the authors to include the source of the data used in this study within a Supplemental Data section that can be accessed and interpreted by readers.. This may include sites of source data or, if not present at a site, raw data. 

We would appreciate receiving your revised manuscript by Feb 24 2020 11:59PM. To enhance the reproducibility of your results, we recommend that if applicable you deposit your laboratory protocols in protocols.io, where a protocol can be assigned its own identifier (DOI) such that it can be cited independently in the future. For instructions see: http://journals.plos.org/plosone/s/submission-guidelines#loc-laboratory-protocols

We look forward to receiving your revised manuscript.

Kind regards,

Arthur J. Lustig, PhD

Academic Editor

PLOS ONE

Journal Requirements:

'All subjects provided written consent to participate in the CCSS, and each participating institution obtained approval to conduct this research through their institutional IRB. The study was conducted in accordance with the Declarations of Helsinki.'

3. In the ethics statement in the manuscript and in the online submission form, please provide additional information about the patient records/samples used in your retrospective study.

Specifically, please ensure that you have discussed whether all data/samples were fully anonymized before you accessed them and/or whether the IRB or ethics committee waived the requirement for informed consent.

If patients provided informed written consent to have data/samples from their medical records used in research, please include this information.

4. Please ensure that your Methods section contains an extensive description of the techniques used for the study, including commercial reagents used for telomere length measurement.

6. Thank you for stating the following in the Competing Interests section:

'I have read the journal's policy and the authors of this manuscript have the following competing interests: Geraldine Aubert is a paid employee of Repeat Diagnostics, Inc., which conducted the telomere length assessment. '

Reviewers' comments:

Reviewer's Responses to Questions

**Comments to the Author**

1. Is the manuscript technically sound, and do the data support the conclusions?

Reviewer #1: Yes

Reviewer #2: Yes

2. Has the statistical analysis been performed appropriately and rigorously? 

Reviewer #1: Yes

Reviewer #2: Yes

3. Have the authors made all data underlying the findings in their manuscript fully available?

Reviewer #1: Yes

Reviewer #2: No

4. Is the manuscript presented in an intelligible fashion and written in standard English?

Reviewer #1: Yes

Reviewer #2: Yes

5. Review Comments to the Author

Reviewer #1: I was specifically asked to review the statistical procedures in the paper “Genetic variation in POT1 and risk of thyroid subsequent malignant neoplasm: A report from the Childhood Cancer Survivor Study”

I find the statistical approaches to be sophisticated and appropriate for the types of analyses discussed. Appropriate methods and corrections have been used and p-values address clearly delineated null hypotheses.

Reviewer #2: Overall the manuscript is well written. The study design is appropriate and the authors address the short comings (the low frequency of the variant limiting gene-environment interactions). This manuscript is a small report but presents a new perspective a on the potential involvement of the POT1 gene that has been previously been show to be linked to a variety of cancers.

6. PLOS authors have the option to publish the peer review history of their article (what does this mean?). If published, this will include your full peer review and any attached files.

Reviewer #1: No

Reviewer #2: No

---

## [Author Response · Author response to Decision Letter 0]

16 Jan 2020

January 14, 2020

Editorial Board

PLOS One

Re: Research Article submission

Dear Editorial Board,

Thank you for reviewing our manuscript and for your comments. We have addressed the request for revisions as follows:

• Query: We need the authors to include the source of the data used in this study 

• Response: Genotype data for the Childhood Cancer Survivor Study are available through dbGaP accession phs001327.v1.p1. I have amended the Data Availability Statement to reflect the accession number, as these data have already been deposited.

In addition, we have addressed recommendations with respect to the PLOS ONE style requirements noted in the decision letter. Specifically: 

• With respect to the request for modification of the Ethics Statement to include the full names of all participating institutional review boards (IRBs), there are 26 institutions who participate in CCSS, and hence 26 separate IRBs who approved the CCSS study. We respectfully request to leave this statement unchanged, in line with a prior CCSS cohort publication by Yang et al. that was accepted to PLOS One, PMID 25764003. If the journal preference is to include the names of all 26 IRBs, we will gladly provide this information.

• We have re-named the figure file to adhere to journal policy

• We have modified the Ethics Statement to provide additional information about the patient records/samples that were used as follows: “All subjects provided written consent to participate in the CCSS, and each of the 26 participating institutions obtained approval to conduct this research through their institutional IRB. Diagnosis and treatment data were initially abstracted from the medical record by the participating treating institution and submitted, fully anonymized, to CCSS. The work described in this study utilized fully anonymized data. The study was conducted in accordance with the Declarations of Helsinki.” 

• The Methods section has been modified to include details on reagents used for telomere length assessment.

• By providing the accession number for these data, we have clarified our statement that anonymized genotype data are publicly available.

• For the request regarding the Competing Interests section, we can affirm that the employment of Dr. Aubert does not alter our adherence to PLOS ONE policies on sharing data and materials. Thank you for your willingness to change the online submission on our behalf. 

Thank you again for your consideration of this work for publication as a Research Article in PLOS One.

Sincerely,

M. Monica Gramatges, MD, PhD

Associate Professor of Pediatrics

---

## [Editor Report · Decision Letter 1]

27 Jan 2020

Genetic variation in POT1 and risk of thyroid subsequent malignant neoplasm: A report from the Childhood Cancer Survivor Study

PONE-D-19-33962R1

Dear Dr. Gramatges,

We are pleased to inform you that your manuscript has been judged scientifically suitable for publication and will be formally accepted for publication once it complies with all outstanding technical requirements.

With kind regards,

Arthur J. Lustig, PhD

Academic Editor

PLOS ONE
---

## [Editor Report · Acceptance letter]

28 Jan 2020

PONE-D-19-33962R1 

Genetic variation in *POT1* and risk of thyroid subsequent malignant neoplasm: A report from the Childhood Cancer Survivor Study 

Dear Dr. Gramatges:

I am pleased to inform you that your manuscript has been deemed suitable for publication in PLOS ONE. Congratulations! Your manuscript is now with our production department. 

With kind regards,

on behalf of

Dr. Arthur J. Lustig 

Academic Editor

PLOS ONE